

# Snow cover variability across glaciers in Nordenskiöldland (Svalbard) from point measurements in 2014–2016

Marco Möller[1, 2, 3] and Rebecca Möller[3, 4]

[1]Institute of Geography, University of Bremen, Bremen, Germany
[2]Geography Department, Humboldt-Universität zu Berlin, Berlin, Germany
[3]Department of Geography, RWTH Aachen University, Aachen, Germany
[4]Geological Institute, Energy and Minerals Resources Group, RWTH Aachen University, Aachen, Germany

*Correspondence to:* Marco Möller (marco.moeller@uni-bremen.de)

**Abstract.** Snow depths and bulk densities of the annual snow layer were measured at 69 different locations on glaciers across Nordenskiöldland, Svalbard, during the spring seasons of the period 2014–2016. Sampling locations lie along nine transects extending over 17 individual glaciers. Several of the locations were visited repeatedly, leading to a total of 109 point measure-
ments, on which we report in this study. Snow water equivalents were calculated for each point measurement. In the dataset,
snow depth and density measurements are accompanied by appropriate uncertainties which are rigorously transferred to the calculated snow water equivalents using a straightforward Monte Carlo simulation-style procedure. The final dataset can be downloaded from the Pangaea data repository (https://www.pangaea.de; doi:10.1594/PANGAEA.896581). Snow cover data indicate a general and statistically significant increase of snow depths and water equivalents with terrain elevation. A signifi-
cant increase of both quantities with decreasing distance towards the east coast of Nordenskiöldland is also evident, but shows
distinct interannual variability. Snow density does not show any characteristic spatial pattern.

## 1   Introduction

Snow cover data are an important fundamentum for various types of studies in cryospheric or climate sciences. The determina-
tion of snow water equivalents is an essential means for quantifying accumulation on glaciers. Together with snow density data
snow water equivalents play a major role as in situ reference especially in spatially distributed mass balance modeling studies
(e.g., Luce et al., 1999; Möller et al., 2011a, 2016b). They also form an important ground truth for various kinds of remote
sensing applications (e.g., Shi et al., 2016; Tedesco et al., 2014) or provide inevitable data bases for terrain climate related
snow re-distribution studies (e.g., Dadic et al., 2010; Lehning et al., 2008).

   Due to their strong relevance for application in calibration or validation procedures, the acquisition of in situ snow cover
data is mostly limited to specific sites being in the focus of glaciological or other snow accumulation-related research. In this
respect, two types of snow cover data exist. First, is the spatially distributed type derived from transient snowpacks which
provides information about the spatial pattern within one annual snow layer. Second, is the temporally distributed type derived
from shallow firn cores which provides information about the multi-year temporal variability of a snow cover at a certain
location.





On the European Arctic archipelago Svalbard local and regional-scale spatial distributions of snow cover have been intensively studied in recent decades. Major regional foci were placed on glaciers around Ny-Ålesund, western Spitsbergen (e.g. Bruland et al., 2001, 2004; Buchroithner et al., 2008; Eckerstorfer and Christiansen, 2011; Hawley et al., 2008; López-Moreno et al., 2016; Valt and Salvatori, 2016), on Hansbreen glacier, southern Spitsbergen (e.g. Grabiec et al., 2006; Laska et al., 2016) and on Vestfonna ice cap, Nordaustlandet (Beaudon et al., 2011; Möller et al., 2011b; Sauter et al., 2013). Apart from these focus areas, various other sites scattered across the glacierized areas of the archipelago have been studied, but mostly by means of snow radar measurements (e.g. Grabiec et al., 2011; Hodgkins et al., 2005; Pälli et al., 2002; Sand et al., 2003; Taurisano et al., 2007; van Pelt et al., 2014; Winther et al., 1998). The last concise overview of snow cover studies on Svalbard has been given by Winther et al. (2003).

Snow cover on Svalbard and elsewhere shows a high spatial variability, which is not only elevation dependent (e.g. Möller et al., 2011a) but also clearly influenced by snow drift-related gains and losses on a local scale (Jaedicke and Gauer, 2005; Jaedicke and Sandvik, 2002; Sauter et al., 2013). As snow cover across the mountainous terrain of the archipelago can thus be expected to show a rather complex pattern, measured snow cover data form an extremely valuable and important basis for calibration and validation of all kinds of glacier mass balance or snow melt models. In this respect, special attention also needs be paid to snow density variations. Missing data about this infrequently measured snowpack characteristic often prohibit an accurate model calibration as only literature-derived assumptions on snow densities are used instead.

Despite the widely known necessities pointed out above, point snow cover data which could readily be employed in validations of Svalbard-wide glacier mass balance modeling studies are hardly available. These studies (Aas et al., 2016; Lang et al., 2015; Möller et al., 2016b; Möller and Kohler, 2018; Østby et al., 2017) mostly use multi-year data from a limited number of locations, meaning they are far better validated regarding their temporal variability than regarding their spatial variability. Hence, it is almost impossible to reveal to which degree these studies are able to capture the range of spatial accumulation variability across Svalbard.

We here present precise snow cover data from a large number of sites on several glaciers scattered across Nordenskiöldland, central Spitsbergen, Svalbard (Fig. 1), a region which is so far underrepresented in the cosmos of Svalbard snow cover-related data retrievals. The data were obtained from in situ point measurements and comprise snow depth, bulk snow density and the derived snow water equivalent. Measurements were carried out during three consecutive spring seasons in the period 2014–2016 and were restricted to the respective annual snow layer. As snow depth distributions in general are known to show a certain degree of persistence over time (Schirmer et al., 2011) and as such persistent pattern have also been reported for Svalbard (Jaedicke and Gauer, 2005), our snapshot measurements of snow cover-related quantities give not only insides into the transient states around the sampling dates, but also into general characteristics of spatial distribution patterns of snow across the Svalbard archipelago.





## 2 Study area

The location of the European Arctic archipelago Svalbard between the northern Atlantic and the Arctic Ocean implies competing influences of fundamentally different air masses and ocean waters. From the South warm and humid air comes in, while from the Northeast cold and dry air approaches the archipelago (Svendsen et al., 2002). The western coast experiences warm
waters delivered by the West Spitsbergen Current (Walczowski and Piechura, 2011), while the eastern parts are affected by cold Arctic Ocean currents (Loeng, 1991). In winter times, air mass movements from the Northeast are the most frequent ones (Käsmacher and Schneider, 2011), implying that easterly weather systems form the major moisture source for snowfall albeit depending on sea-ice conditions (Førland et al., 1997; Rogers et al., 2001).

The snowfall pattern resulting from these synoptic-scale forcing is per se influenced by orographic effects making annual
precipitation sums increase with terrain elevation (Hagen et al., 1993). This pattern is superimposed by a regional decrease towards the interior of the archipelago (Humlum, 2002). Owing to the proximity of warm ocean waters that result in high moisture flux during winter {walczowski2011, an additional, negative southwest-northeast gradient exists (Hagen et al., 1993). The resulting vertical gradients of snow accumulation show considerable, regional variability across Svalbard (e.g. Grabiec et al., 2011; Möller et al., 2011b; Taurisano et al., 2007). The snow cover found on Svalbard reflects the specific combination
of competing climate and ocean influences. A typical snowpack shows considerable amounts of depth hoar at the bottom, a number of icy wind crusts and a low overall depth which is rather comparable to continental tundra sites. This set of special snow cover characteristics was termed 'High Arctic maritime snow climate' by Eckerstorfer and Christiansen (2011).

Melt onset occurs in the period May–June, starting in the south and gradually proceeding towards the northeast, while freeze-up tends to start in the centre of the archipelago, spreading to its edges over the period September–October (Sharp and Wang,
2009). The near-surface wind regime is predominantly controlled by katabatic components induced by the large ice masses (e.g. Claremar et al., 2012).

## 3 Data description

### 3.1 Field measurements

Measurements of snow cover characteristics across Nordenskiöldland were performed during three 5 to 9-day periods of
fieldwork carried out between late March and mid April of the period 2014–2016. Measurements were carried out along nine different transects extending across 17 different glaciers (Fig. 1, Tables 1 and 2). The transects were laid out to approximately run along the central flowlines of the glaciers wherever possible in order to avoid more risky terrain. Three of those transects (C, D, E) were visited during all three field seasons. Two transects (G, H) were visited twice and four (A, B, F, I) only once. The transects consisted of four to twelve individual sampling locations each. In total, 109 point measurements were carried out
at 69 different locations (Fig. 2).

At each sampling location, one out of two different sets of measurements was carried out. First, was a full measurement, which consisted of both snow depth and snow density measurements. Second, was a partial measurement, which consisted of



the snow depth measurement only. Choices between the two were made in the field on the basis of time and safety issues, mostly related to weather conditions. Full measurements account for 74 out of the 109 point measurements in total (Fig. 2, Table 1).

Snow depth measurements were performed for the annual snow layer, i.e. down to the last end-of-summer surface, using an avalanche probe. Detection of the last end-of-summer surface by manual sounding is straightforward in Arctic climate settings. Across the low-lying ablation areas, the last end-of-summer surface equals the surface of the glacier ice body. Across the high-lying accumulation areas it is formed by a continuous ice crust within the snowpack that lies below a clearly identifiable layer of autumn hoar. This characteristic succession of a layer of very soft, coarse-crystalline snow underlain by a hard, icy layer is climate induced and has regularly been described before (e.g. Möller et al., 2011b; Steffen et al., 1999).

Bulk snow density was determined over a continuous column of the snowpack. At each sampling location a core was excavated from the annual snow layer using a Kovacs Mark III coring system. Detection of the last end-of-summer surface was again straightforward as the penetration of the autumn hoar layer was easily perceptible during the process of manual coring. The excavated snow core was weighed and the weight then converted into bulk snow density on the basis of known core depth and cross section of the core barrel.

Uncertainty in the snow depth soundings arises from small scale snowpack variability. To account for it, a series of ten snow depth soundings was carried out at each sampling location. The final snow depth at each sampling location was calculated as the mean over the respective series of soundings. The standard deviation over the series of soundings is assumed as measure of uncertainty. Uncertainty in the determination of bulk snow density arises from the potential undercatch of core weight. Loose fresh snow on top of the core or loose autumn hoar at its bottom might get lost during excavation and transfer into the weighing pan. However, this loss affects only minor amounts compared to the entire snow core and a fraction of 5% of the density is considered an adequate and rather conservative measure of uncertainty.

## 3.2 Data preparation

While snow depth was regularly measured, bulk snow density was only determined for 74 of the 109 point measurements. This leaves the task to fill up the missing bulk snow density data with adequate substitutes. As bulk snow density frequently varies with terrain elevation along the transects, missing density data ($\rho_m$) at elevation $z_m$ is inter- or extrapolated from the closest two measurements ($\rho_1$ and $\rho_2$) at terrain elevations $z_1$ and $z_2$ according to:

$$\rho_m = \rho_1 + (z_0 - z_1) \cdot (\rho_2 - \rho_1)/(z_2 - z_1) \tag{1}$$

Uncertainties for the density substitutes are assumed to equal a fraction of 10% of the densities, i.e. twice the originally assumed 5%.

The snow water equivalent is calculated for each point measurement by multiplying snow depth with the proportion between bulk snow density and the density of water ($1000\,\mathrm{kg\,m^{-3}}$). Snow water equivalent has to be accompanied by an adequate measure of uncertainty which has to result from the individual uncertainties assumed for snow depth and density. This is done via a Monte Carlo simulation-style procedure (e.g. Möller et al., 2016a). In order to achieve stable results, 900 individual





calculations were carried out. For each calculation, snow depth and bulk snow density were randomly varied within their respective probability densities. Each of these probability densities is defined by a mean and a standard deviation. The means are given by the respective values of snow depth and bulk snow density. The standard deviations are assumed to equal the respective measures of uncertainty. The final snow water equivalent value is then calculated as the mean over the 900 separately

calculated values. The associated measure of uncertainty of the snow water equivalent is represented by the standard deviation over the same 900 values.

## 4 Results

The mean (± one sigma) snow depth over all 109 point measurements is $1.44\pm0.50$ m. Combined with a mean bulk snow density of $424\pm86\,\mathrm{kg\,m^{-3}}$ this implies a mean snow water equivalent of $0.63\pm0.28$ m w.e. The one sigma spreads indicate that

the variability of snow depth is higher than that of bulk snow density. The former ranges between 0.46 and 2.70 m, the latter between 139 and $598\,\mathrm{kg\,m^{-3}}$. Consequently, the absolute range of snow water equivalents documented in the dataset is even more higher, showing a substantial spread between 0.09 m w.e. at location I-10 (132 m a.s.l.) in 2016 and 1.36 m w.e. at location E-05 (614 m a.s.l.) also in 2016.

The distributions of snow depths and snow water equivalents show evidence for characteristic spatial pattern. Both quantities

exhibit significant positive correlations with terrain elevation and longitude (Tables 3 and 4), indicating increasing snow depths and water equivalents with either increasing terrain elevation or decreasing distance to the eastern coast of Nordenskiöldland. Bulk snow density, in contrast, does only show very limited evidence for any of such pattern.

Snow water equivalent shows considerable interannual variability, but a reliable comparison can only be obtained from the subset of twelve sampling locations at transects C, D and E where measurements were conducted during all three years (Fig.

2). At these locations, 2016 sticks out as the year richest in snow with a mean snow water equivalent of 0.90 m w.e. followed at a considerable distance by 2015 (0.60 m w.e.) and 2014 (0.54 m w.e.).

This interannual variability is not necessarily a result of varying overall snow accumulation but may also be a result of different patterns of spatial snow cover variability in different years. This possibility is indicated by different strengths of the correlation between snow water equivalent and longitude (Table 4). In 2014 and 2016, snow water equivalent shows signifi-

cant (99% level) partial correlations (leaving out influences of terrain elevation) of $r = 0.71$ and $r = 0.76$ with longitude. This documents an increase of snow water equivalent towards the East, which is not present in 2015. As transects D and E, i.e. ten out of twelve locations, are located close to the east coast of Nordenskiöldland (Fig. 1, Table 2), this increase of snow water equivalents automatically implies higher annual mean values in the considered subset of sampling locations than in 2015.

The marked interannual variability of the correlations between snow depth and longitude as well as between snow water

equivalent and longitude can be seen as the most prominent finding of this study. It suggests annually varying patterns of moisture transport towards Nordenskiöldland. These can climatologically be explained by varying frequencies of precipitation-relevant atmospheric circulation patterns during winter (Käsmacher and Schneider, 2011).





## 5  Data availability

The final dataset contains 109 individual sets of point data. Each set shows metadata and snow cover data. The former part consists of a) location ID, b) geographical coordinates, c) altitude, d) sampling date and e) glacier ID in the Randolph Glacier Inventory 6.0 (Pfeffer et al., 2014). The latter part consists of a) snow depth together with its uncertainty, b) bulk snow density together with its uncertainty and a flag identifying whether density was measured or inter-/extrapolated and c) snow water equivalent together with its uncertainty. It is available for download (https://www.pangaea.de; doi:10.1594/PANGAEA.896581) from the Pangaea earth and environmental sciences data repository (Möller and Möller, 2018).

## 6  Summary

Snow cover data were obtained from 109 individual point measurements. For 74 of these measurements snow depth was obtained together with bulk snow density. The former was determined by repeated soundings with an avalanche probe. The latter was derived by weighing a snow core which was previously extracted from the annual snow layer using a Kovacs Mark III coring system. For the remaining 35 point measurements only snow depth was determined directly, while snow density was interpolated or extrapolated from neighboring sampling locations. Snow water equivalents were calculated for all 109 cases from snow depth and bulk snow density, accounting for their individual uncertainties by using a Monte Carlo simulation-style procedure. Thus, snow water equivalents are accompanied by rigorous uncertainty estimates. Measurements were conducted along nine different transects that extend across 17 individual glaciers scattered across Nordenskiöldland. They represent the late March to mid April field situations of the period 2014–2016.

The compiled dataset shows a mean snow depth of $1.44\pm0.50\,\mathrm{m}$, a mean bulk snow density of $424\pm86\,\mathrm{kg\,m^{-3}}$ and a mean snow water equivalent of $0.63\pm0.28\,\mathrm{m}$ w.e. The variability of snow water equivalent between individual point measurements is far higher than the variability of bulk snow density or snow depth. The former as well as the latter show a characteristic variability with terrain elevation indicated by a statistically significant, positive partial correlation (i.e. excluding influences of longitudinal variability). The dataset also reveals a characteristic longitudinal variability in the form of increasing snow depths and water equivalents towards the east coast of Nordenskiöldland. However, this spatial pattern is not persistent and only occurs during 2014 and 2016. This suggests annually varying spatial precipitation patterns across the study area.

*Author contributions.*  MM designed and conducted fieldwork and data compilation activities, wrote the manuscript and prepared all figures. RM was crucially involved in fieldwork and data analysis and reviewed the manuscript.

*Competing interests.*  The authors declare that they have no conflict of interest.

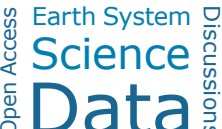

*Acknowledgements.* This study and the related data acquisition was funded by grant nos. MO2653/1-1 and MO2653/1-2 of the German Research Foundation (DFG). The authors are grateful to all people involved into the fieldwork activities on Svalbard.





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

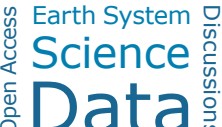



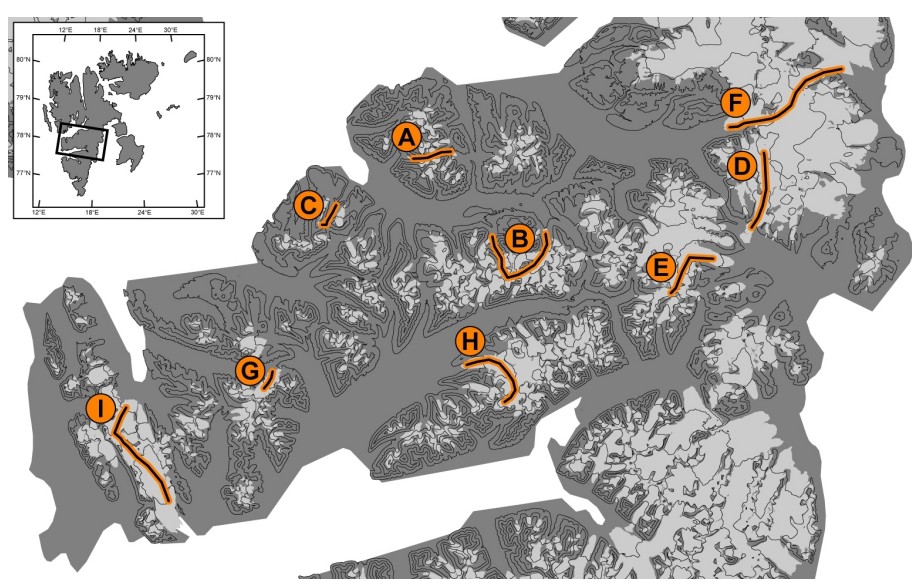

**Figure 1.** Overview map of Nordenskiöldland and its glaciers. The greyscale identifies land areas (dark grey) and glacierized areas (light grey). The locations of the nine measurement transects A to I (cf. Table 2 and Fig. 2 for details) are indicated in orange. Contour spacing on the map is 200 m.



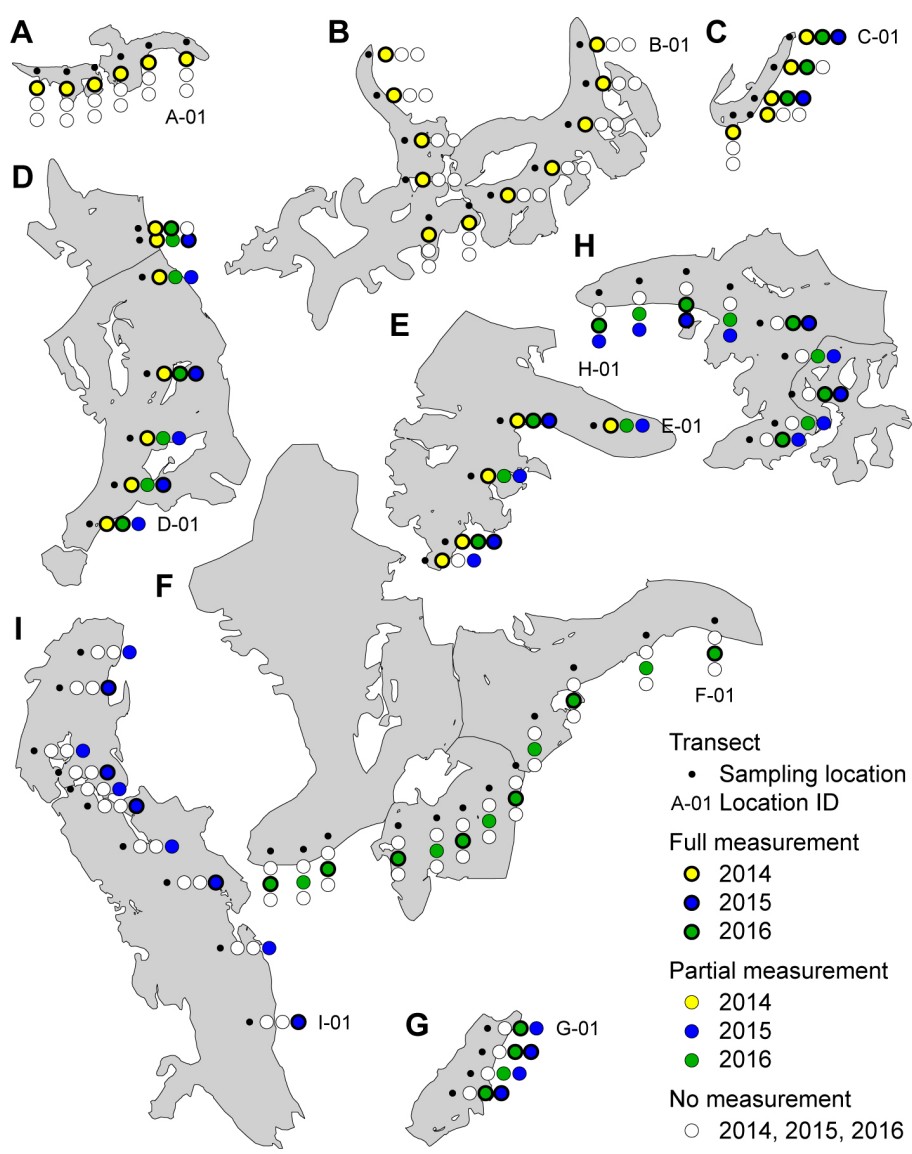

**Figure 2.** Detailed maps of the nine measurement transects. The transects (indicated by their capital letter names) are drawn on top of the glaciers they cover. For each transect the sampling locations are shown together with colored symbols indicating if full or partial measurements were conducted in the respective years. Location IDs are only shown at the start of each transect for reasons of clarity.



**Table 1.** Overview of fieldwork activities for snow cover sampling. Transects refer to Figures 1 and 2. Point measurements comprising both snow depth and density (full measurements) are distinguished from point measurements comprising only snow depth (partial measurements).

| Quantity | 2014 | 2015 | 2016 |
|---|---|---|---|
| Transects | A, B, C, D, E | C, D, E, F, G, H | C, D, E, G, H, I |
| Dates | March 29 – April 6 | March 24 – March 30 | April 5 – April 9 |
| Full measurements | 34 | 23 | 17 |
| Partial measurements | 0 | 16 | 19 |



**Table 2.** Overview on the measurement transects. The location IDs of the sampling points are given together with the ranges of latitudes, longitudes and terrain elevations covered by each transect. Glaciers that are covered by the respective transect are given with their ID in the Randolph Glacier Inventory 6.0 (Pfeffer et al., 2014). The years in which each transect was visited are given in addition.

| Transect | IDs | Latitudes (°) | Longitudes (°) | Elevations (m a.s.l.) | Glaciers | Years |
|---|---|---|---|---|---|---|
| A | A-01–A-06 | 78.247–78.256 | 16.058–16.247 | 339–631 | 07.01114, 07.01116 | 2014 |
| B | B-01–B-11 | 78.093–78.147 | 16.494–16.830 | 235–754 | 07.01475, 07.01130, 07.01567 | 2014 |
| C | C-01–C-05 | 78.165–78.189 | 15.438–15.527 | 283–676 | 07.01107 | 2014, 2015, 2016 |
| D | D-01–D-07 | 78.145–78.238 | 18.101–18.204 | 101–701 | 07.00428, 07.00427 | 2014, 2015, 2016 |
| E | E-01–E-05 | 78.067–78.110 | 17.582–17.852 | 199–614 | 07.00409 | 2014, 2015, 2016 |
| F | F-01–F-12 | 78.274–78.339 | 17.996–18.716 | 57–550 | 07.01476, 07.01480, 07.01478 | 2015 |
| G | G-01–G-04 | 77.957–77.978 | 15.077–15.130 | 260–410 | 07.01071 | 2015, 2016 |
| H | H-01–H-09 | 77.935–77.990 | 16.317–16.608 | 202–832 | 07.00344, 07.00342 | 2015, 2016 |
| I | I-01–I-10 | 77.813–77.931 | 14.169–14.501 | 132–586 | 07.01100, 07.01082 | 2016 |

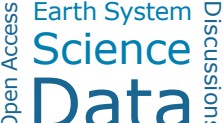

**Table 3.** Correlations and partial correlations (in parenthesis) between different snow cover parameters and terrain elevation. Partial correlations exclude influences of longitude (cf. Table 4). Bold values are significant on the 99% level.

| Period | Snow depth | Bulk snow density | Snow water equivalent |
|---|---|---|---|
| All data | **0.449** (**0.526**) | 0.220 (**0.258**) | **0.402** (**0.469**) |
| 2014 | **0.522** (**0.820**) | 0.160 (0.195) | **0.519** (**0.744**) |
| 2015 | **0.527** (**0.530**) | **0.414** (**0.428**) | **0.522** (**0.526**) |
| 2016 | **0.517** (**0.565**) | **0.304** (0.243) | **0.459** (**0.489**) |

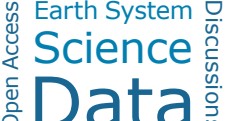

**Table 4.** Correlations and partial correlations (in parenthesis) between different snow cover parameters and longitude. Partial correlations exclude influences of terrain elevation (cf. Table 3). Bold values are significant on the 99% level.

| Period | Snow depth | Bulk snow density | Snow water equivalent |
|---|---|---|---|
| All data | **0.351** (**0.454**) | 0.246 (**0.280**) | **0.328** (**0.412**) |
| 2014 | **0.527** (**0.821**) | 0.095 (0.148) | **0.439** (**0.711**) |
| 2015 | -0.041 (0.074) | 0.028 (0.123) | -0.031 (0.085) |
| 2016 | **0.729** (**0.751**) | **0.627** (**0.609**) | **0.749** (**0.760**) |