# Peer review of "Snow cover variability across glaciers in Nordenskiöldland (Svalbard) from point measurements in 2014–2016"

_Earth System Science Data, 2018_

## Referee Comment (RC1) · Anonymous Referee #1 · 29 Jan 2019

Snow data is important in climate studies of the cryosphere components. Time series are needed as input and validation of remote sensing and modelling data. The authors present data from a region in Svalbard were few (but some) snow data is available before. In that context this paper provides some useful data. The paper is in general well written and easy to follow. They have an extensive reference list. However, I have some problems with the applications of the data they present. They say that snow data is needed as validation in Svalbard-wide glacier mass balance modelling studies, but that requires that the data is representative for the snow cover and thus can be used in validation. I am doubtful that this is the case with the data they present. The survey was done early in the spring season, late March and early April. A consider-

able amount of snow can be accumulated after this period, in April/May, often up to 30 % of the total winter snow fall. Only three of the glaciers have been measured all three years 2014, 2015 and 2016, glaciers C, D and E. Two glaciers, A and B, were only measured in 2014, F only in 2015 and I only in 2016. Over the three years they measure altogether 109 points. In the results they give the average of all these 109 as a result with a mean of 0,63 m w.e. But that is in my opinion not useful and is just a number saying that there is fairly low snow accumulation in this region of Svalbard. Especially since the number of 109 samplings is taken from different glaciers and different years. They state that there is a substantial spread in the snow depth and water equivalent. That is fairly obvious and when they select ten points on a glacier for snow probing they may easily hit a place that is not at all representative for that elevation band since it could be a place with wind-drift and almost now snow accumulation. This is well known from all who run mass balance programs. Therefore snows probing on glaciers for mass balance usually have a large number of points spread out to cover the spatial variability. Their main finding is probably that there is large spatial and inter-annual variability in snow precipitation, but then a one year sampling from one glacier has very limited value for general information about the snow cover and as validation data. They could also have discussed their finding in relation to precipitation data from the Norwegian Meteorological Institute synoptic weather station in Longyearbyen. Is there any correlation? Or they could have compared their data with the mass balance data, winter accumulation data, from the monitoring program by the Norwegian Polar Institute from the Kongsfjorden area. Ground Penetrating Data could have been done simultaneously and would have extended the spatial distribution and indicated representativeness of the point data. Fig. 2. I do not understand why they put in marks (open circles) for years without any measurements. It only gives an impression that there are more measurements than it actually is. On four glaciers, Glaciers A, B I and F only one year of measurements are done. I think they should simply delete all the open circles. Then it would be easier to see where and which year they did sampling.

---

## Referee Comment (RC2) · Anonymous Referee #2 · 6 Feb 2019

General Comments:

This paper reports on in-situ ("glaciological") measurements of snow depth and density in 3 spring seasons across 17 glaciers in south-central Spitzbergen. The survey transects sample elevation gradients, while the study glaciers span a range of longitudes. Numerous local measurements of snow depth using avalanche probes were averaged to obtain the reported values, while densities were measured less frequently by weighting snow cores. Depth and density are combined to estimate SWE and uncertainties assigned using a MC procedure.

The dataset is useful and the paper generally clear and complete. The main concern

[Figure]

I would highlight is that the method of density estimation at unmeasured sites implicitly assumes an elevation dependence that is apparently unsupported by the data. I recommend the authors undertake an alternative approach for estimating density (see comments) and report on the differences between their chosen method and one that does not assume elevation-dependence. A minor concern is that there are several instances where claims are made in the paper, seemingly on the basis of common knowledge and/or experience, that should be fortified with references and/or evidence. Finally, the paper would benefit from one or two additional figures illustrating the dependence of measured/estimated quantities on terrain parameters.

Specific Comments (by section or page.line):

Abstract: Is there some way to make the relationship between the following combination of numbers more clear: 109 point measurements, 69 locations, 9 transects, 17 glaciers? For example, it is not clear to me how many measurements or transects per glacier were made. 9 transects and 17 glaciers makes it wound like 8 glaciers were not measured which is clearly not the case. Were all 9 transects executed on each glacier? Please clarify in the abstract so the reader does not have to wait for Figure 1 to understand the survey design.

1.23-24. The second type (temporal) could also be derived by means other than firn cores. What about repeat measurements of Type 1 over multiple years?

2.15-16. "Missing data about this infrequently measured snowpack characteristic often prohibit an accurate model calibration". Would be nice to add a reference for this.

2.23. Omit "precise". Precision can only be determined by testing, so it seems odd to label the data as precise at this point.

4. 5. "Detection of the last end-of-summer surface by manual sounding is straightforward in Arctic climate settings." Is there a reference and/or more information that could be added? It is hard to imagine that there are never fall rain or thaw events that would

create ice layers at depth in the ablation area. Are ice layers never found in pits/cores, or are they so thin that they are never misinterpreted as the glacier surface?

4.10-14. Coring procedure. Some more detail would be appreciated by this reader at least who has encountered challenges in making these very same measurements. What were the air temperatures at the time of coring? Were cores always easily recoverable? Were chips always unambiguously able to be separated from cores? How is uncertainty in core mass assessed? Was core mass/density ever compared to snow mass/density as assessed in snow pits or with other coring devices (e.g. SWE tube)?

4.15. Uncertainty in the snow depth soundings arises from small scale snowpack variability => arises in part from

4.15-17. Over what footprint (area) were the 10 measurements made? Given that there are multiple scales of variability in snow-water equivalent, it seems important to define the scale at which uncertainty is assessed.

4.18. "Uncertainty in the determination of bulk snow density arises from the potential undercatch of core weight." This is certainly easy to imagine, but has the core-measured density ever been compared to other density measurements? It is also easy to imagine chips not being entirely separated from the core, and thus contributing to overestimation of density. Can this effect be ruled out?

4.20-21. " However, this loss affects only minor amounts compared to the entire snow core and a fraction of 5% of the density is considered an adequate and rather conservative measure of uncertainty." Please provide a reference or some supporting evidence. This uncertainty assessment seems optimistic.

4.24. " fill up the missing bulk snow density data with adequate substitutes" => estimate snow density where measurements are missing

4.24-25. " As bulk snow density frequently varies with terrain elevation along the transects." In the authors' experience? In general? Please specify. There are examples

where snow density does not vary with elevation. Was this procedure for density assignment tested, e.g., against values of measured density? Is two values the right number? Why not try a fixed search radius instead (with a minimum number of measurements required) so as not to unnecessarily mix spatial scales in different density assignments?

4.33. 900 MC simulations per point value of SWE?

5.8-13. Please also report means and std for the subset of data where density was measured. The values reported include both measured and estimated densities (and therefore SWE values).

5.17. " Bulk snow density, in contrast, does only show very limited evidence for any of such pattern." This observation would appear to undermine the approach taken to estimate densities at the unmeasured locations. Suggest an alternative approach to density estimation using a spatial search radius and no elevation dependence. This calculation should at least be attempted and the differences between it and the current approach reported (or, better yet, included in the uncertainty assessment).

5.20-21. Please include uncertainties on these values. It is not clear whether the values from 2015 and 2014 are indistinguishable.

6.19-20. " The variability of snow water equivalent between individual point measurements is far higher than the variability of bulk snow density or snow depth." This seems true almost by definition, unless variation in depth and density somehow cancelled in the determination of SWE. It would be more informative in the conclusion just to report that snow depth was found to be more variable than density.

Figures. It would be really nice to have some figures that showed the relationship (or lack thereof) between snow depth, density, SWE and elevation, longitude, rather than just the correlations reported in the tables.

Technical (by page.line):

1.16. "inevitable". Wrong word, but not clear what the correct word would be. "invaluable"?

1.20-24. Grammatically, the last two sentences are not quite right. Should be "The first is spatially distributed and derived from . . ."

2.28. pattern have => patterns have

2.29. insides => insights

3.9. these synoptic-scale forcing => forcings

3.12. {walczowski2011: citation not compiled properly

3.15. " at the bottom" redundant with depth hoar

3.29. carried out => made

5.12. delete "more"

5.20-21. "2016 sticks out as the year richest in snow" => stands out as the year with the most snow.

6.15. " Thus, snow water equivalents are accompanied by rigorous uncertainty estimates." Delete. The sentence before speaks for itself.

Specific questions in review instructions:

Are the data and methods presented new? Data are new, methods are not.

Is there any potential of the data being useful in the future? Yes.

Are methods and materials described in sufficient detail? For the most part. See detailed comments above.

Are any references/citations to other data sets or articles missing or inappropriate? Not to my knowledge.

Is the article itself appropriate to support the publication of a data set? Yes.

Is the data set accessible via the given identifier? Yes

Is the data set complete? Appears to be

Are error estimates and sources of errors given (and discussed in the article)? Yes

Are the accuracy, calibration, processing, etc. state of the art? They are standard

Are common standards used for comparison? N/A

Is the data set significant – unique, useful, and complete? Yes

Consider article and data set: Are there any inconsistencies within these, implausible assertions or data, or noticeable problems which would suggest the data are erroneous (or worse). No. See suggestions about density estimation.

Is the data set itself of high quality? Yes

Is the data set usable in its current format and size? Are the formal metadata appropriate? Yes

Is the length of the article appropriate? Yes

Is the overall structure of the article well structured and clear? Yes

Is the language consistent and precise? Yes. Article would benefit from a final proof-reading by a native speaker.

Are mathematical formulae, symbols, abbreviations, and units correctly defined and used? N/A

Are figures and tables correct and of high quality? Yes, though I recommend adding one or two figures.

Is the data set publication, as submitted, of high quality? Yes

---

## Referee Comment (RC3) · Anonymous Referee #3 · 6 Mar 2019

This manuscript describes data collection and processing for 109 point snow depth/density measurements on 17 different glaciers in Svalbard over three years.

Major Comments:

1. Although this is a large dataset and it is clear that distributed snow depth/density observations have significant value for a number of different scientific fields, it's unclear to me whether this specific dataset, given its spatiotemporal sampling, can be fully utilized. While acknowledging that this is a ESSD submission, the manuscript and corresponding dataset would be much stronger if it included additional analyses (i.e., comparison to a reanalysis product or other in situ weather/glaciological observations)

[Figure]

that demonstrate the datasets value.

2. A potentially different and valuable direction would be to emphasize the snow density observations, rather than the depth/SWE observations. Seventy-four density observations is actually a huge dataset (often similar snow studies have 3-8 density observations), and thus the finding that density shows limited variability is quite useful and could result in a highly-cited paper.

3.These observations were collected during the late winter/early spring, so presumably more snow accumulated between the sampling date and peak accumulation. Constraining this additional accumulation would increase the datasets value, as it would align these observations with a glaciological mass balance framework. In addition, the Study Area section should have a brief description about these three winters – were they normal? What was unique about them? Such information would prove useful for interpreting this dataset and the observed interannual variability.

4. Although the Monte Carlo simulations are one approach for estimating density uncertainty, it seems like a potentially more insightful approach would be to calculate the predicted density using Equation 1 for all known density locations (using neighboring sites) and then reporting the RMSE between the observed and predicted.

5. The manuscript notes that the most prominent finding is the "marked interannual variability of the correlations between snow depth and longitude". Adding scatterplots of swe/density vs. elevation/longitude would be a revealing and useful addition to the manuscript.

Minor Comments: 1-16 – what is meant by inevitable data bases? 1-17 – the meaning of terrain climate is unclear, perhaps for "studies looking at snow redistribution due to the interaction between terrain and meteorological forcings." 2-10 – Is this referring to glacierized terrain specifically or just terrestrial snow covered terrain? 2-11 – replace "snow drift related gains and losses" with wind redistribution 2-12 - replace "can thus be expected to show a rather" with exhibit a complex pattern. . .. 2-13 – replace "snow

cover data form" with snow observations are an... 2-17 – replace the "widely known necessities" with Despite the clear importance of snow observations, such datasets are lacking and thus limit the validation of glacier mass balance studies. 2-23 – what is meant by precise snow cover? 2-24 – remove "in the cosmos" 2-29 – replace "insides" with insight 4-16 – soundings suggest some sort of sonic or sonar approach, could "probings" be used instead. 5-8 – reporting the mean snow depth across three years and across a range of transects doesn't actually seem all the meaningful. 5-14 – replace "show evidence for" with exhibit and patterns rather than pattern 5-18- replace "but" with thus 5-20 – replace "sticks out as the year richest in snow" with had the greatest snow depths

---

## Author Comment (AC1) · 14 Apr 2019

**Initial statement**

We thank the three reviewers for their thorough work and valuable comments. We intend to prepare a revised manuscript that will account for all the issues raised by the reviewers and that will thus vastly benefit from their suggestions.

According to several comments made on issues regarding density data and especially following a strong suggestion of reviewer 3 we intend to shift the major analytical focus of the manuscript from snow water equivalents to snow density. The reviewer highlighted in his comment (RC3-2) that we present an extraordinarily numerous dataset of bulk snow density measurements that is far beyond the number that is usually presented and used in comparable studies. Going along with this shift of the major analytical focus, we will also retitle the revised manuscript as follows: "Limited bulk density variation in snow covers across glaciers in central Svalbard". We agree with reviewer 3 that with this new focus the manuscript will not only present some new point snow data that could be used in model calibration and validation, but it will emphasize an important finding (the limited density variation). This finding might be known already in the community, but it has so far never been analyzed and outlined explicitly because of (as also stated by the reviewer) a lack of extensive spatially distributed density data. With our revised manuscript, we intend to fill this gap in referenceable documentations of snow cover characteristics.

As part of this new focus, the revised manuscript will roughly show the following changes:

- The introduction section will be slightly altered in order to explicitly outline the usage of snow density data in mass balance modeling studies.
- The applied methods will be extended by including alternative ways of estimating the bulk snow densities at points where only snow depth was measured. This was explicitly requested by reviewers 2 (RC2-12, RC2-15) and 3 (RC3-4). In addition to the already used interpolation between neighboring data points we will apply different spatial search radii as suggested by reviewer 2. Both approaches will also be applied to data points with existing density data in order to facilitate an appropriate uncertainty discussion as suggested by reviewer 3.
- The results section will be rearranged and extended regarding snow density issues in order to more directly account for the new focus of the manuscript.

Apart from these overall changes we will account for the specific comments made by the reviewers in full detail as outlined below.

**Comments by reviewer 1**

RC1-1: However, I have some problems with the applications of the data they present. They say that snow data is needed as validation in Svalbard-wide glacier mass balance modelling studies, but that requires that the data is representative for the snow cover and thus can be used in validation. I am doubtful that this is the case with the data they present. The survey was done early in the spring season, late March and early April. A considerable amount of snow can be accumulated after this period, in April/May, often up to 30 % of the total winter snow fall.

*Answer: The globally accepted and applied standard in glaciological mass balance measurements is formed by measuring both ablation and accumulation at stakes which are installed at fixed locations on the glacier (e.g. Cogley et al. 2011). The repeated stake readings are used either for direct calculations of glacier wide balances by extrapolating the measurements over the area-altitude distribution of the glacier or by using them for calibration and validation of glacier mass balance models. In any case these data are point measurements which by no means can be guaranteed to be representative. And exactly because of this, it is extremely valuable for mass balance modeling studies to be able to rely on as many point data*

*as possible. Taking this into account, our data represent the standard form of point data of accumulation on a glacier. They even go beyond the usually applied stake readings in terms of accuracy, as we explicitly measured bulk density of the snow layer at the majority of the measurement locations. Regarding the timing of our surveys (early spring) the reviewer is right, that substantial snow fall will be added to the snow layer before the ablation season starts. However, as we intend our data to be used for calibration and validation purposes of mass balance models this does not matter at all. The vast majority of these models runs on daily resolution, so there is no need for specific dates to be used regarding the reference data.*

RC1-2: Only three of the glaciers have been measured all three years 2014, 2015 and 2016, glaciers C, D and E. Two glaciers, A and B, were only measured in 2014, F only in 2015 and I only in 2016. Over the three years they measure altogether 109 points. In the results they give the average of all these 109 as a result with a mean of 0,63 m w.e. But that is in my opinion not useful and is just a number saying that there is fairly low snow accumulation in this region of Svalbard. Especially since the number of 109 samplings is taken from different glaciers and different years.

*Answer: The reviewer is right that overall mean values are not useful to report and will be thus be deleted from the revised version of the manuscript.*

RC1-3: They state that there is a substantial spread in the snow depth and water equivalent. That is fairly obvious and when they select ten points on a glacier for snow probing they may easily hit a place that is not at all representative for that elevation band since it could be a place with wind-drift and almost now snow accumulation. This is well known from all who run mass balance programs. Therefore snows probing on glaciers for mass balance usually have a large number of points spread out to cover the spatial variability.

*Answer: As stated above in our answer to RC1-1, the common standard for mass balance measurements are stake readings. Extensive snow probing across the full spatial variability of a glacier is rarely done and mostly limited to few intensively studied reference glaciers. Hence, the reviewer is right with referring to "mass balance programs". However, on newly or rarely studied glaciers taking point data from along the flowline is preferred due to logistical limitations and safety issues (crevasses at the margins etc.). In order to overcome influences of small scale snow depth variability (influenced by the micro-topography of the ice surface, e.g. by meltwater channels), we did extensive snow depth probing around each sampling point (as already stated in the manuscript). However, we see the point of the reviewer, that more extensive point measurements (in terms of spatial variability) would be the ideal way of measuring accumulation and that our (standard) way of doing it along the flowline only will not cover the full range of spatial variability of snow coverage. Accounting for this, the revised manuscript will contain a passage that outlines explicitly that our measurements are only point measurements at selected locations that does not intend to be fully representative for a certain part of the glacier (an issue that could anyway barely be achieved in any kind of measured environment-related data).*

RC1-4: Their main finding is probably that there is large spatial and interannual variability in snow precipitation, but then a one year sampling from one glacier has very limited value for general information about the snow cover and as validation data. They could also have discussed their finding in relation to precipitation data from the Norwegian Meteorological Institute synoptic weather station in Longyearbyen. Is there any correlation?

*Answer: The reviewer raised an interesting issue that we would like to follow in the revised manuscript. We will add snow depth data from the Met Norway station at Longyearbyen airport as reference. Further we will discuss our own measurements of snow depth on the glaciers in central Spitsbergen in relation to these reference data. This will yield additional insights into potential inter-annual variations of the longitudinal snow depth pattern.*

RC1-5: Or they could have compared their data with the mass balance data, winter accumulation data, from the monitoring program by the Norwegian Polar Institute from the Kongsfjorden area. Ground Penetrating Data could have been done simultaneously and would have extended the spatial distribution and indicated representativeness of the point data.

*Answer: Comparing our data with mass balance data from the Kongsfjorden area would not yield any usable new insights. We concentrate on Nordenskiöldland, i.e. the central part of Spitsbergen, in our study. Data from a region that is situated substantially further north would not be comparable to ours. Using ground penetrating radar (GPR) would of course have been a fantastic addition to our data. But there is always more that can be done in a study and using GPR was out of scope in our study. Apart from that, GPR only delivers snow depth information. Snow density has still to be measured manually in order to transfer GPR data into snow water equivalent information.*

RC1-6: Fig. 2. I do not understand why they put in marks (open circles) for years without any measurements. It only gives an impression that there are more measurements than it actually is. On four glaciers, Glaciers A, B I and F only one year of measurements are done. I think they should simply delete all the open circles. Then it would be easier to see where and which year they did sampling.

*Answer: The reviewer is completely right here and the figure will be changed accordingly for incorporation in the revised manuscript.*

**Comments by reviewer 2**

RC2-1: Abstract: Is there some way to make the relationship between the following combination of numbers more clear: 109 point measurements, 69 locations, 9 transects, 17 glaciers? For example, it is not clear to me how many measurements or transects per glacier were made. 9 transects and 17 glaciers makes it wound like 8 glaciers were not measured which is clearly not the case. Were all 9 transects executed on each glacier? Please clarify in the abstract so the reader does not have to wait for Figure 1 to understand the survey design.

*Answer: We thank the reviewer for unveiling this apparent shortcoming of the abstract. We will revise the related text passages accordingly. In the revised manuscript the text will be clear about the numbers, i.e. that measurements were done at 69 locations along nine transects which extend over 17 glaciers. As some transects were measured repeatedly a total number of 109 point measurements was done.*

RC2-2: 1.23-24. The second type (temporal) could also be derived by means other than firn cores. What about repeat measurements of Type 1 over multiple years?

*Answer: The reviewer is right. Temporally distributed data could also be derived by repeat measurements of type 1. This information will be added to the respective text passage of the revised manuscript.*

RC2-3: 2.15-16. "Missing data about this infrequently measured snowpack characteristic often prohibit an accurate model calibration". Would be nice to add a reference for this.

*Answer: Unfortunately, there is, to our knowledge, no specific reference for this statement. It is a widely known fact that by far not all mass balance measurements are accompanied by bulk density measurements done at the same locations. Instead, either a limited number of*

*density measurements is treated as representative for a certain area or density data are even taken from the literature only. In the revised manuscript we focus more on the density data as explicitly suggested by reviewer 3 (cf. our answer to comment RC3-2 and our initial statement). Along with that we will provide a text passage in the new introduction that deals with common handling of density data in mass balance modeling studies. In this passage we will give specific references to different ways of treatment of density information and the related implications.*

RC2-4: 2.23. Omit "precise". Precision can only be determined by testing, so it seems odd to label the data as precise at this point.

*Answer: The revised manuscript will be corrected accordingly.*

RC2-5: 4.5. "Detection of the last end-of-summer surface by manual sounding is straightforward in Arctic climate settings." Is there a reference and/or more information that could be added? It is hard to imagine that there are never fall rain or thaw events that would create ice layers at depth in the ablation area. Are ice layers never found in pits/cores, or are they so thin that they are never misinterpreted as the glacier surface?

*Answer: We see the point of the reviewer and are thankful for unveiling this shortcoming in the formulation. Of course there are even winter rain events that create ice layers in the snowpack. However, the critical feature identifying the end-of-summer surface is the pronounced layer of autumn hoar on top of this hard surface. When extracting the cores, it is easy to drill through the rain-event ice layers before at some point the core barrel instantaneously breaks through the autumn hoar layer, indicating that you've reached the end-of-summer surface. Hence, any ice layers within the snowpack can be identified in the extracted core and taken into account during snow depth sounding. This will be outlined in more detail in the revised manuscript.*

RC2-6: 4.10-14. Coring procedure. Some more detail would be appreciated by this reader at least who has encountered challenges in making these very same measurements. What were the air temperatures at the time of coring? Were cores always easily recoverable? Were chips always unambiguously able to be separated from cores? How is uncertainty in core mass assessed? Was core mass/density ever compared to snow mass/density as assessed in snow pits or with other coring devices (e.g. SWE tube)?

*Answer: This comment and the one before (RC2-5) suggest, that additional details are needed regarding the coring procedure. We will therefore provide a largely extended text passage related to this issue and to the field measurements in general in the revised manuscript. Coring was carried out at air temperatures roughly between -5 and -15°C. Exact measurements were not carried out. However, the process of coring was found to be independent of any meteorological issues at the times it was carried out. Cores were always easily recoverable, even if drilling in higher density snow packs was certainly harder than in low density ones. We did no traditional snow pit for comparison, as it can be assumed that a bulk snow density from a core barrel is superior in terms of accuracy. During density measurements in a snow pit wall it is rather easy that material of the snow column gets lost between the individual measurement steps. A shortcoming that does not apply to a continuous core. Hence, there is no need to compare the core density to any snow pit derived "traditional" density measurements.*

RC2-7: 4.15. Uncertainty in the snow depth soundings arises from small scale snowpack variability => arises in part from

*Answer: Will be changed accordingly in the revised manuscript.*

RC2-8: 4.15-17. Over what footprint (area) were the 10 measurements made? Given that there are multiple scales of variability in snow-water equivalent, it seems important to define the scale at which uncertainty is assessed.

*Answer: The ten measurements were made within a 20-30 meter circle around the sampling point. This will also be explained in the extended passage on field measurements (cf. answer to RC2-6).*

RC2-9: 4.18. "Uncertainty in the determination of bulk snow density arises from the potential undercatch of core weight." This is certainly easy to imagine, but has the core measured density ever been compared to other density measurements? It is also easy to imagine chips not being entirely separated from the core, and thus contributing to overestimation of density. Can this effect be ruled out?

*Answer: Regarding the comparison to of core-derived bulk snow density to densities derived from other types of measurements see our answer to comment RC2-6. Regarding the issue of separation of snow chips from the core barrel, the reviewer is right that more information needs to be added to the manuscript. After each coring process and final extraction of the snow core from the core barrel, the amounts of snow chips remaining inside the barrel were always very small, but were extracted as good as possible under field conditions and added to the weighing pan before measurement. Nevertheless, uncertainty related to this issue remains. The respective text passage of the revised manuscript will be changed in order to also address this specific issue.*

RC2-10: 4.20-21. " However, this loss affects only minor amounts compared to the entire snow core and a fraction of 5% of the density is considered an adequate and rather conservative measure of uncertainty." Please provide a reference or some supporting evidence. This uncertainty assessment seems optimistic.

*Answer: There is no reference that supports this assumption. However, a short calculation might do so. Assuming a core length of 100 cm with about 10 cm autumn hoar included (which is quite a high estimate) and a bulk density of 400 kg/m³ for the 90 cm of snow and 200 kg/m³ for the 10 cm of hoar, even the loss of the entire autumn hoar part of the core would result in an underestimation of just 5.26%. Hence, our estimate of 5% can be seen as appropriate.*

RC2-11: 4.24. " fill up the missing bulk snow density data with adequate substitutes" => estimate snow density where measurements are missing

*Answer: Will be changed accordingly in the revised manuscript.*

RC2-12: 4.24-25. " As bulk snow density frequently varies with terrain elevation along the transects." In the authors' experience? In general? Please specify. There are examples where snow density does not vary with elevation. Was this procedure for density assignment tested, e.g., against values of measured density? Is two values the right number? Why not try a fixed search radius instead (with a minimum number of measurements required) so as not to unnecessarily mix spatial scales in different density assignments?

*Answer: This sentence seemingly created a misunderstanding. It was not meant that there is any systematic variation (increase or decrease) with elevation, but only variation along the transect. This will be corrected in the revised manuscript. Regarding the procedure of density assignment to the unmeasured sampling points we refer to our initial statement and also to our answer to comments RC2-15 and RC3-4.*

RC2-13: 4.33. 900 MC simulations per point value of SWE?

*Answer: Yes, this is right. MC simulations need a huge number of runs until they deliver stable results. However, it only sounds much, in terms of computational time it is still nothing mentionable in case of such a simple calculation.*

RC2-14: 5.8-13. Please also report means and std for the subset of data where density was measured. The values reported include both measured and estimated densities (and therefore SWE values).

*Answer: We will follow the request of the reviewer and include these numbers in the revised manuscript.*

RC2-15: 5.17. " Bulk snow density, in contrast, does only show very limited evidence for any of such pattern." This observation would appear to undermine the approach taken to estimate densities at the unmeasured locations. Suggest an alternative approach to density estimation using a spatial search radius and no elevation dependence. This calculation should at least be attempted and the differences between it and the current approach reported (or, better yet, included in the uncertainty assessment).

*Answer: The issue of density estimation for the unmeasured points will get much more and detailed attention in the revised manuscript, as it was also raised in several other reviewer comments (e.g. RC2-12, RC3-4). For details we refer to our initial statement.*

RC2-16: 5.20-21. Please include uncertainties on these values. It is not clear whether the values from 2015 and 2014 are indistinguishable.

*Answer: Uncertainty measures will be included in the revised manuscript.*

RC2-17: 6.19-20. " The variability of snow water equivalent between individual point measurements is far higher than the variability of bulk snow density or snow depth." This seems true almost by definition, unless variation in depth and density somehow cancelled in the determination of SWE. It would be more informative in the conclusion just to report that snow depth was found to be more variable than density.

*Answer: The reviewer is right with his statement and the respective text passage in the summary section will be changed accordingly in the revised manuscript.*

RC2-18: Figures. It would be really nice to have some figures that showed the relationship (or lack thereof) between snow depth, density, SWE and elevation, longitude, rather than just the correlations reported in the tables.

*Answer: We thank the reviewer for this very valuable idea. A related figure with six different panels (annually color-coded scatter plots between a) elevation and snow depth, b) elevation and density, c) elevation and SWE, d) longitude and snow depth, e) longitude and density and f) longitude and SWE) will be added to the revised manuscript.*

RC2-19: Technical comments

*Answer: The technical comments made by the reviewer are greatly appreciated and the revised manuscript will be changed/corrected according to the suggestions made.*

**Comments by reviewer 3**

RC3-1: Although this is a large dataset and it is clear that distributed snow depth/density observations have significant value for a number of different scientific fields, it's unclear to me whether this specific dataset, given its spatiotemporal sampling, can be fully utilized. While acknowledging that this is a ESSD submission, the manuscript and corresponding dataset would be much stronger if it included additional analyses (i.e., comparison to a reanalysis product or other in situ weather/glaciological observations) that demonstrate the datasets value.

*Answer: The reviewer is right that additional analyses would strengthen the manuscript. However, we so far refrained from including such kind of work because on the website of ESSD it is explicitly stated that "Any interpretation of data is outside the scope of regular articles." (/about/manuscript_types.html). Adding comparisons to other types of data would not yield any insightful results if we would not interpret the deviations or similarities between those different types of data. Nevertheless, we will include snow depth data from the Met Norway weather station at Longyearbyen airport as a reference (cf. answer to comment RC1-4). By doing so we will be able to present our measured data in relation to a fixed and standardized measurement taken at an official weather station. This would shed more light on the issue of the spatial variability of snow depth which was raised implicitly in several reviewer comments and would thus help to demonstrate the datasets value.*

RC3-2: A potentially different and valuable direction would be to emphasize the snow density observations, rather than the depth/SWE observations. Seventy-four density observations is actually a huge dataset (often similar snow studies have 3-8 density observations), and thus the finding that density shows limited variability is quite useful and could result in a highly-cited paper.

*Answer: We appreciate this suggestion very much and would like to follow it in the revised manuscript. Explicit details on this issue are given in our initial statement.*

RC3-3: These observations were collected during the late winter/early spring, so presumably more snow accumulated between the sampling date and peak accumulation. Constraining this additional accumulation would increase the datasets value, as it would align these observations with a glaciological mass balance framework. In addition, the Study Area section should have a brief description about these three winters – were they normal? What was unique about them? Such information would prove useful for interpreting this dataset and the observed interannual variability.

*Answer: Constraining any additional accumulation after the date of measurement would mean to apply/develop existing/new methods of downscaling solid precipitation from the synoptic situation to small-scale terrain variability, including considerations of wind redistribution. However, this is a research field of its own and could not be readily applied as suggested by the reviewer. As we intend our data to be applied in mass balance model calibration the timing of sampling is not a shortcoming at all (cf. our answer to RC1-1), as such models in the meantime mostly run on daily resolutions they are easily able to handle validation data that do not cover entire seasons. Especially as logistical limitations virtually always prohibit that glaciological measurements are exactly done at peak-accumulation or peak-ablation dates. Regarding the requested information about the three winters, we will add a paragraph dealing with climatic information to the revised manuscript.*

RC3-4: Although the Monte Carlo simulations are one approach for estimating density uncertainty, it seems like a potentially more insightful approach would be to calculate the predicted density using Equation 1 for all known density locations (using neighboring sites) and then reporting the RMSE between the observed and predicted.

*Answer: As already stated in our answer to comment RC2-12 and RC2-15 the issue of density estimation for the unmeasured points will get much more and detailed attention in the revised manuscript. For details we refer to our initial statement.*

RC3-5: The manuscript notes that the most prominent finding is the "marked interannual variability of the correlations between snow depth and longitude". Adding scatterplots of swe/density vs. elevation/longitude would be a revealing and useful addition to the manuscript.

*Answer: Such scatterplots will be added to the revised manuscript as this issue has also been raised by reviewer 2 (cf. our detailed answer to comment RC2-18).*

RC3-6: 1-16 – what is meant by inevitable data bases?

*Answer: The reviewer is right, this is confusing. We will change the respective passage from "inevitable data bases" to "necessary reference data" as this formulation gets much more to the point.*

RC3-7: 1-17 – the meaning of terrain climate is unclear, perhaps for "studies looking at snow redistribution due to the interaction between terrain and meteorological forcings."

*Answer: The correct word would have been "topoclimate". The respective text passage will be changed in the revised manuscript to avoid further misunderstandings.*

RC3-8: 2-10 – Is this referring to glacierized terrain specifically or just terrestrial snow covered terrain?

*Answer: Its actually true for both. Snow depth generally increases with terrain elevation but is modulated on a local scale by wind-drift related gains and losses of snow.*

RC3-9: 2-23 – what is meant by precise snow cover?

*Answer: The word "precise" will be deleted in the revised manuscript.*

RC3-10: 4-16 – soundings suggest some sort of sonic or sonar approach, could "probings" be used instead.

*Answer: This is a valuable suggestion and we will follow it in the revised manuscript. All mentions of "snow depth soundings" will be replaced by "snow depth probings".*

RC3-11: 5-8 – reporting the mean snow depth across three years and across a range of transects doesn't actually seem all the meaningful.

*Answer: The reviewer is right and the same issue has already been raised by reviewer 1 (cf. our answer to comment RC1-2). Hence, we will not mention such values in the revised manuscript.*

---

## Editor Comment (EC1) · Reinhard Drews (Editor) · 15 Apr 2019

Dear Authors and Reviewers,

thank you all for submitting your constructive comments and the respective responses. Based on the criticism raised, I discourage submission of a revised version at this stage.

Details of this decision are as follows:

Reviewers 1 & 2 gave differing recommendations on how to proceed with this discussion paper, which is why I invited a third reviewer. All reviewers agree that the data

[Figure]

presented here are unique and also potentially useful for other researches. (I add to this that related field work was surely not an easy task.) However, particularly reviewer 1 and 3, question application of the data as currently presented (RC1-1 // RC3-1) and both don't recommend publication in ESSD. The authors recognize parts of their criticism and suggest to "shift the major analytical focus" towards the snow density variability (RC3-3). I think this can be a promising direction, however, together with the other proposed changes (incl. of new Figures, incl. new data from weather stations, alternative interpolation schemes,..) this will introduce a new scientific topic and go beyond a normal revision. I therefore follow the suggestions from reviewer 1 an 3, but hope that the authors can incorporate the raised comments and publish the data in a new paper.

Kind regards, Reinhard Drews
* * *